# Reliability Study of an Intelligent Profiling Progressive Automatic Glue Cutter Based on the Improved FMECA Method

**Heng Zhang [1], Yaya Chen [1], Jingyu Cong [1,2], Junxiao Liu [3,4], Zhifu Zhang [3] and Xirui Zhang [1,3,4,\*]**

1   School of Information and Communication Engineering, Hainan University, Haikou 570228, China
2   State Key Laboratory of Marine Resource Utilization in South China Sea, Hainan University, Haikou 570228, China
3   School of Mechanical and Electrical Engineering, Hainan University, Haikou 570228, China
4   Sanya Nanfan Research Institute, Hainan University, Sanya 572025, China
\*   Correspondence: zhangxr@hainanu.edu.cn

**Abstract:** This study introduces the fuzzy theory approach as an enhancement to the traditional failure mode, effect, and criticality analysis (FMECA) method in order to address its limitations, which primarily stem from subjectivity and a lack of quantitative analysis. The proposed method, referred to as FMECA improvement based on fuzzy comprehensive evaluation, aims to quantify the qualitative aspect of the analysis and provides a detailed outline of the analysis procedure. By applying the enhanced FMECA method to assess the reliability of an intelligent profiling progressive automatic rubber cutter, the hazard ranking for each failure mode of the cutter can be determined, thereby identifying areas that require reliability improvement. The analysis outcomes demonstrate that this method establishes a theoretical foundation for subsequent cutter improvement designs, enables early identification of potential failures, and consequently leads to a reduced failure rate and an enhanced reliability level for the intelligent profiling progressive automatic cutter. Furthermore, this innovative agricultural equipment reliability analysis and testing approach holds significant value in elevating the reliability standards of agricultural equipment as a whole and can be explored and implemented in other agricultural machinery contexts.

**Keywords:** natural rubber; intelligent profiling progressive automatic glue cutter; FMECA method; fuzzy comprehensive evaluation; reliability analysis



## 1. Introduction

The intelligent profiling step-by-step automatic rubber cutting machine offers several advantages, including addressing the shortage of rubber workers, revolutionizing the rubber work process, high automation levels, and independence from environmental constraints. These benefits effectively reduce the reliance on manual labor for rubber cutting, lower labor costs, and increase natural rubber output. Moreover, the machine's reliability directly influences the yield and quality of natural rubber. Currently, rubber cutting machines experience a significant failure rate, which greatly diminishes the quality of the cut rubber and can even damage the rubber trees, thereby adversely affecting the income of rubber farmers. Therefore, it is imperative to conduct a reliability analysis of rubber cutting machines. Failure is a disruptive event that can cause production delays and compromise the overall reliability of a system [1–3]. In order to cope with various failure modes that may occur, appropriate decisions based on the multicriteria decision-making (MCDM) process are made at different stages such as design, manufacturing, and operation to improve system reliability [4].

FMECA (failure modes, effect, and criticality analysis) mainly consists of two parts: failure mode and effect analysis (FMEA) and criticality analysis (CA). It is commonly used to find and solve various known or potential failures in equipment systems, which plays a crucial role in improving the reliability and service life of equipment. However, when

using the traditional FMECA method to analyze the reliability of equipment, there is too much subjectivity, when determining the order of hazard level only qualitative analysis can be performed (not quantitative analysis) and it is difficult to find out the weak links in the system accurately by calculating the objective results, and it cannot provide technical support for the daily maintenance of various equipment systems. In recent decades, significant efforts have been made by scholars and researchers to enhance the FMECA methodology [5]. Several improved methods have emerged, focusing on the following areas. Bozdag et al. proposed a novel fuzzy FMECA method based on fuzzy sets [6]. This approach considered the optimal weights of risk factors and integrated them using an ordered weighted average operator based on the cut concept. Liu et al. [7] introduced fuzzy directed graph and matrix methods into FMECA, developing a new FMECA model that considered the relative weights of risk factors expressed linguistically. These weights were transformed into fuzzy numbers and risk priority indices for failure modes were calculated using corresponding fuzzy risk matrices. Zhou et al. [8] proposed a generalized evidence FMECA model (GEFMECA) to handle uncertain risk factors encompassing both conventional and incomplete risk factors. By utilizing the generalized evidence theory, the issue of relative weights among risk factors was effectively resolved. Additionally, Liu et al. [9] introduced an integrated FMECA method based on interval intuitionistic fuzzy sets (IVIFS) and multi-attribute boundary approximation region comparison (MABAC) methods. This approach established a linear programming model for obtaining weight information among risk factors when complete weight information was not available, thereby determining the optimal weights for these factors. Yang et al. [10] proposed a fuzzy rule-based Bayesian inference method for prioritizing failure modes. Jee et al. [11] introduced a new fuzzy inference system (FIS)-based risk assessment model for FMECA, employing a two-stage approach to reduce the collection of fuzzy rules. Gajanand et al. [12] combined FMECA with a fuzzy linguistic scaling method, presenting a novel reliability-centered maintenance strategy that utilizes weighted Euclidean distance and fuzzy logic-based center-of-mass defuzzification to rank failure modes. Sayyadi et al. [13] developed a new FMECA model based on an intuitionistic fuzzy approach, enabling the evaluation of failure modes in the presence of fuzzy concepts and limited data. Jian et al. [14] combined intuitionistic fuzzy sets (IFSS) with evidence theory, proposing a novel FMECA failure mode risk assessment method. Jiang et al. [15] utilized fuzzy affiliation in the proposed FMECA fuzzy evidence method to evaluate the risk factors of failure modes, ranking them using the Dempster–Shafer evidence theory to consolidate characteristic information. Aydogan [16] proposed a method that combines the rough hierarchical analysis and fuzzy TOPSIS methods for organizational performance analysis in a fuzzy environment. Liu et al. [17] introduced an intuitive fuzzy hybrid TOPSIS method to determine the risk priority of failure modes in FMECA. Silvia et al. [18] proposed an optimization method for the maintenance activities of complex systems by integrating reliability analysis and MCDM methods. They used *AHP* for weight assessment and fuzzy TOPSIS for risk prioritization. Zhou et al. [19] introduced gray theory and fuzzy theory into FMECA for tanker equipment failure prediction. They determined the risk priority of failure modes using the fuzzy risk priority number (FRPN) obtained from fuzzy set theory and the gray correlation coefficient from gray theory. Liu et al. [20] developed an FMECA framework by integrating the cloud model and PROMETHEE method to handle diverse risk assessments from FMECA team members and prioritize failure mode risks. Mandal et al. [21] proposed a method to rank human errors using the VIKOR method. Baloch et al. [22] integrated the fuzzy VIKOR method and data envelopment analysis into FMECA to determine the ranking of potential modalities and select the most important damage modalities. Liu et al. [23] proposed an improved approach for FMECA using fuzzy evidential reasoning (FER) theory. This approach addresses two limitations of traditional FMECA: the acquisition and aggregation of evaluations from different experts and the determination of the risk priorities for failure modes. Liu et al. [24] further proposed a new FMECA failure mode prioritization risk assessment model based on FER and belief rule base (BRB) methods. In this model,

the diverse and uncertain evaluations provided by experts are captured and aggregated using FER, whereas the nonlinear and uncertain relationships between risk factors and corresponding risk levels are modeled using the BRB. Du et al. [25] presented a fuzzy FMECA method based on evidential reasoning (ER) and TOPSIS to accurately identify and aggregate risk factors. Su et al. [26] proposed an improved method for dealing with conflicting evidence combinations by employing an uncertain inference method based on the Gaussian distribution to reconstruct the basic belief assignment (BBA) while considering the weight of each expert. Jiang et al. [27] proposed an improved application of the theory of evidence to FMECA, which redistributes the underlying belief assignments by considering the credibility coefficients obtained based on the distance of the evidence to minimize conflicts between expert opinions. The applied research conducted by these experts and scholars, focusing on enhancing the FMECA method in the reliability analysis of different equipment systems, establishes a theoretical foundation for analyzing the reliability of the intelligent profiling progressive automatic rubber cutter discussed in this paper.

This paper focuses on conducting reliability research on an intelligent profiling progressive automatic rubber cutter. The research utilizes an improved FMECA method as the research tool. The primary objective is to assess and enhance the reliability level of the rubber cutter. The application of the improved FMECA methodology innovatively combines expert knowledge and engineering experience to provide a quantitative analysis solution for the reliability of rubber cutting equipment through improved fuzzy theory. The study aims to identify key areas for improving the reliability of the rubber cutter. The findings of this research will provide theoretical guidance for subsequent improvement designs and the daily maintenance of rubber cutters.

## 2. Materials and Methods

### 2.1. Basic Theory of the Traditional FMECA Method

The fundamental principle of the traditional FMECA is to systematically examine the structure of the system, identify potential failure modes, analyze the underlying causes of failures, and use statistical methods to estimate the severity ($S$), occurrence ($O$), and detection ($D$) of the failure consequences based on technical specifications, historical data, and user requirements. Subsequently, the risk priority number (RPN) is calculated, and the failure modes are prioritized according to their RPN values. Appropriate improvement or maintenance measures can then implemented to reduce the RPN and ensure the reliability of the system.

The typical steps involved in conducting FMECA analysis are as follows, as illustrated in Figure 1:

(1) Product Definition: Provide a description of the product's composition, environmental conditions during operation, functional aspects, and operational procedures;

(2) FMECA Method Selection: Choose the appropriate FMECA method based on the analysis objective and the product's development stage and develop a corresponding FMECA analysis table;

(3) FMEA Analysis Implementation: Identify potential failure modes, describe their effects, investigate the contributing causes for each mode, assess failure detection methods, and analyze potential compensatory measures;

(4) Hazard Analysis: Evaluate the hazards associated with the identified failure modes.

In traditional FMECA, each failure mode identified within a system is evaluated using three risk factors: severity ($S$), occurrence ($O$), and detectability ($D$). The RPN is calculated by multiplying the values of S, O, and D, providing a ranking for the failure modes [28]. Typically, experts assign scores ranging from 1 to 10 to the risk factors S, O, and D, with higher values indicating more severe situations. The RPN value is utilized to determine the risk priority of each failure mode, allowing analysts to identify inherent vulnerabilities in the system. A higher RPN indicates greater importance [29], suggesting a more significant impact on the system and a higher risk priority. To ensure safety and reliability, preventive

and improvement measures should give priority to higher-risk failure modes in order to prevent their occurrence.

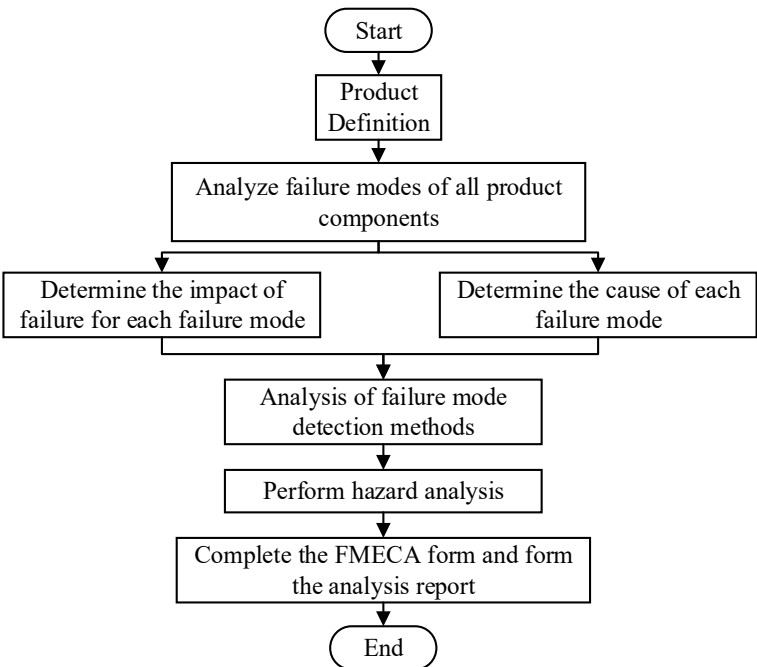

**Figure 1.** The basic analysis steps of the traditional FMECA method.

However, the conventional crisp value of the RPN has faced criticism for several reasons, [30–35] as explained below:

1. The relative importance of the three risk factors is often not considered or assumed to be equal, which may not reflect the actual situation in many cases;

2. Multiplying the values of S, O, and D across different groups may result in the same RPN values but the risk hazards in each group could be completely different. This can lead to an unreasonable allocation of limited resources and time, or worse, the neglect of high-risk failure modes;

3. Calculating the mean RPN is sensitive and contentious when the values of the risk factors change. Even a slight change in the value of a risk factor can have a significantly different effect on the RPN, particularly when other risk factor values are large;

4. The evaluation of risk factors S, O, and D usually utilizes discrete ordered metric scales. As a result, the multiplicative calculations lack meaningfulness since the resulting RPN may show discontinuity, with multiple gaps and a wide range from 1 to 1000. In such situations, the ranking results of failure modes lose their significance and can potentially mislead;

5. The accurate determination of these three risk factors is often challenging. FMECA team members express their evaluations using linguistic terms such as high, medium, or low;

6. Due to variations in expertise and backgrounds, FMECA team members may evaluate the same risk factors differently and some evaluations may be ambiguous and uncertain. The traditional FMECA approach lacks comprehensive methods to describe group judgments and explore the inherent connections between different evaluations. [36].

Although the FMECA method facilitates the timely identification of structural design flaws, comparison of alternative solutions, and decision-making support for improving design and maintenance strategies, the analysis process poses challenges due to multiple evaluation factors, qualitative assessments, and the incommensurability of failure consequences and impacts. As a result, analysts often encounter difficulties in producing precise and effective analysis results.

### 2.2. Enhancing the Fundamentals of the FMECA Method

The fuzzy comprehensive evaluation method is a quantitative approach based on fuzzy mathematics that converts qualitative assessments into quantitative evaluations using fuzzy principles. It facilitates the overall evaluation of objects or phenomena influenced by multiple factors, providing clear and systematic results. This method is especially valuable for addressing problems involving fuzziness and difficulties in quantification, making it suitable for a range of non-deterministic scenarios.

The enhanced FMECA method integrates fuzzy theory with the traditional FMECA method to analyze the reliability of equipment systems. By leveraging the strengths of both reliability analysis methods, it effectively addresses the limitations of the traditional FMECA approach. The improved method allows for the quantification of analysis results and enhances the accuracy and reliability of comprehensive hazard-level rankings for each failure mode. Figure 2 illustrates the fundamental evaluation process of the enhanced FMECA method.

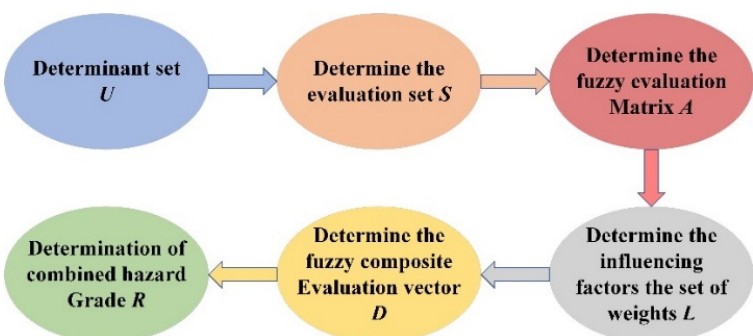

**Figure 2.** The fundamental evaluation process of the enhanced FMECA method.

#### 2.2.1. Defining the Set of Factors

The factor set, represented by $U$, encompasses the collection of factors that exert influence on the evaluation object.

$$U = \{u_1, u_2, u_3, \ldots, u_i, \ldots, u_m\} \tag{1}$$

where $u_i$ denotes the $i$th influencing factor, $i = 1, 2, 3, \ldots, m$.

#### 2.2.2. Determining the Evaluation Set

Defining the evaluation set, represented by $S$, includes all the evaluation results provided by experts for the evaluated object.

$$S = \{s_1, s_2, s_3, \ldots, s_j, \ldots, s_n\} \tag{2}$$

where, $s_j$ denotes the $j$th evaluation result made by the judging expert, $j = 1, 2, 3, \ldots, n$.

#### 2.2.3. Establishing the Fuzzy Evaluation Matrix

A fuzzy mapping $f: U \to S$ is needed to establish the relationship between the evaluation results and the influence factors. This mapping is denoted as $f: U \to F(S)$, where each influence factor $u_i$ is mapped to its respective fuzzy evaluation result $f(u_i)$. Applying this mapping allows us to determine the degree of affiliation $a_{ij}$ between an influence factor $u_i$ and an evaluation result $s_j$.

$$f(u_i) = \mathbf{A}_i = (a_{i1}, a_{i2}, a_{i3}, \ldots, a_{ij}, \ldots, a_{in}) \tag{3}$$

The set $\mathbf{A}_i$ represents the evaluation results for the individual influence factor, $u_i$. These individual evaluation results are organized as rows in the evaluation matrix and

then transposed to form an *m-by-n* matrix. The resulting matrix, denoted as **A**, represents the fuzzy evaluation of the influence factors.

$$
\mathbf{A} = [\mathbf{A}_1, \mathbf{A}_2, \cdots, \mathbf{A}_n]^{\mathrm{T}} = \begin{bmatrix} a_{11} & a_{12} & \cdots & a_{1n} \\ a_{21} & a_{22} & \cdots & a_{2n} \\ \vdots & \vdots & \vdots & \vdots \\ a_{m1} & a_{m2} & \cdots & a_{mn} \end{bmatrix} \tag{4}
$$

An evaluation team, comprising *r* experts, is formed, and each expert is responsible for providing their evaluation result $s_j$ for each influence factor $u_i$. If there are $r_{ij}$ experts who evaluate $u_i$ as $s_j$, the evaluation set for $u_i$ can be obtained using the following procedure:

$$
\begin{aligned}
\mathbf{A}_i &= \left\{ \frac{r_{i1}}{r}, \frac{r_{i2}}{r}, \frac{r_{i3}}{r}, \cdots, \frac{r_{ij}}{r}, \cdots, \frac{r_{in}}{r} \right\} \\
&= \{a_{i1}, a_{i2}, a_{i3}, \cdots, a_{ij}, \cdots, a_{in}\}
\end{aligned} \tag{5}
$$

where $\sum\limits_{j=1}^{n} a_{ij} = 1$.

### 2.2.4. Determine the Weights for the Set of Influence Factors

Given the significant variations in the degree of harm caused by different influencing factors on each failure mode, it is crucial to prioritize and rank the factors according to their impact. Prior to conducting the comprehensive evaluation by experts, the weights are determined in a specific order. These weights collectively constitute the weight set of influencing factors, denoted as **L**.

$$
\mathbf{L} = \{l_1, l_2, l_3, \cdots, l_i, \cdots, l_n\} \tag{6}
$$

where $0 < l_i < 1$, $i \in [1, n]$, and $\sum\limits_{i=1}^{n} l_i = 1$.

Several methods are available for determining the weight set of influencing factors, including survey statistics, expert scoring methods, and the analytic hierarchy process (AHP). Among these methods, the survey statistics method involves analyzing a significant amount of data through empirical research to determine the weights. However, this method is labor intensive and time consuming, making it impractical in many situations. The expert scoring method determines the weights through expert evaluations. Although this method saves time and effort, it is subjective and prone to errors, posing challenges in obtaining the ideal and precise weights. In contrast, the analytic hierarchy process method can address the limitations of the expert scoring method by mitigating human biases in the weight determination process. This method ensures the attainment of objective and effective weights. In this study, the 1–9 scale model of the analytic hierarchy process is used to determine the weight set of influencing factors [8]. The following steps outline the procedure:

(1) The relative significance of the influencing factors $u_i$ and $u_j$ is conveyed by using $b_{ij}$, resulting in the formation of a judgment matrix **B**.

$$
\mathbf{B} = \begin{bmatrix} b_{11} & b_{12} & \cdots & b_{1n} \\ b_{21} & b_{22} & \cdots & b_{2n} \\ \vdots & \vdots & \vdots & \vdots \\ b_{n1} & b_{n2} & \cdots & b_{nn} \end{bmatrix} \tag{7}
$$

where the values of $b_i$ are referred to in Table 1. It is evident that $b_{ii} = 1$ and $b_{ij} = 1/b_{ji}$;

**Table 1.** AHP analysis of the relative importance degree of the influencing factors.

| Implications | $b_{ij}$ |
|---|---|
| $u_i$ is as important as $u_j$ | 1 |
| $u_i$ is slightly more important than $u_j$ | 3 |
| $u_i$ is significantly more important than $u_j$ | 5 |
| $u_i$ is strongly more important than $u_j$ | 7 |
| $u_i$ is definitely more important than $u_j$ | 9 |
| The importance of $u_i$ over $u_j$ is between the above two scale values | 2, 4, 6, 8 |

(2) A consistency test is conducted on the judgment matrix **B** by determining the consistency ratio $K_C$. To begin, the maximum characteristic root $\lambda_{\max}$ associated with the judgment matrix **B** and the consistency index $I_C$ are calculated as follows:

$$I_C = \frac{\lambda_{\max} - n}{n - 1} \tag{8}$$

Subsequently, the value of $I_T$, representing the average random consistency index of the judgment matrix, is determined. The specific values of $I_T$ for the judgment matrices of order 1–13 can be obtained from Table 2.

**Table 2.** Values 1–13 of $I_T$ judgment matrix.

| $n$ | 1 | 2 | 3 | 4 | 5 | 6 | 7 | 8 | 9 | 10 | 11 | 12 | 13 |
|---|---|---|---|---|---|---|---|---|---|---|---|---|---|
| $I_T$ | 0 | 0 | 0.58 | 0.90 | 1.12 | 1.24 | 1.32 | 1.41 | 1.45 | 1.49 | 1.52 | 1.54 | 1.56 |

Based on the calculated values of $I_C$ and $I_T$, the consistency ratio $K_C$ is computed, which is defined as follows:

$$K_C = \frac{I_C}{I_T} \tag{9}$$

when $K_C < 0.1$, it can be concluded that the consistency of judgment matrix **B** meets the required criteria. However, if $K_C$ exceeds 0.1, adjustments to the judgment matrix **B** are necessary;

(3) The weights of each factor are determined using the square root method. The weighting term for the $i$th factor of the analyzed failure mode is calculated as follows:

$$l_i = \frac{\sqrt[n]{\prod_{j=1}^{n} b_{ij}}}{\sum_{i=1}^{n} \sqrt[n]{\prod_{j=1}^{n} b_{ij}}} \tag{10}$$

Subsequently, the set of factor weights for this specific failure mode is obtained as $\mathbf{L}_1 = \{l_1, l_2, \ldots, l_i, \ldots, l_n\}$ where $0 < l_i < 1$, ensuring the fulfillment of the normalization condition:

$$\sum_{i=1}^{n} l_i = 1 \tag{11}$$

### 2.2.5. The Calculation of the Fuzzy Comprehensive Evaluation Vector

By multiplying the previously derived fuzzy evaluation matrix **A** of influence factors with the set of influence factor weights **L**, we can obtain the fuzzy comprehensive evaluation vector, denoted as **D**, which can be expressed as follows:

$$\mathbf{D} = \mathbf{L} \times \mathbf{A} = [l_1, l_2, \cdots, l_n] \times \begin{bmatrix} a_{11} & a_{12} & \cdots & a_{1n} \\ a_{21} & a_{22} & \cdots & a_{2n} \\ \vdots & \vdots & \vdots & \vdots \\ a_{m1} & a_{m2} & \cdots & a_{mn} \end{bmatrix} \tag{12}$$

### 2.2.6. Determining the Comprehensive Hazard Level

To enable a clearer comparison of the hazard levels associated with each failure mode in relation to the evaluation object, it is essential to calculate a specific value by applying weighted averaging to the fuzzy comprehensive evaluation vector **D**. This value, represented as **R**, indicates the ranking of the hazard levels for each failure mode with respect to the evaluation object.

$$\mathbf{R} = \mathbf{D} \cdot \mathbf{S}^{\mathrm{T}} \tag{13}$$

where **S** in denotes the evaluation result matrix.

## 3. Results and Analysis

The quality of rubber cutting plays a crucial role in achieving high yields from natural rubber trees in agricultural production. Therefore, the performance and reliability of rubber cutters directly affect the quality of rubber cutting, necessitating a reliability analysis. This study aims to analyze the reliability of a specific type of rubber cutter, specifically an intelligent profiling progressive automatic rubber cutter, as an illustrative example.

### 3.1. Intelligent Profiling Progressive Automatic Gum Cutter

The structure of the intelligent profiling progressive automatic rubber cutter, as depicted in Figure 3, mainly comprises several key components, including the adjustable flexible tooth-belt fixing device, adaptive tree profiling cutting device, circumferential motion device, vertical motion device, reduction drive device, and a complete set of screws, motors, and other auxiliary elements.

Before operating the intelligent profiling automatic rubber cutter, careful considerations are taken into account for the variations in the upper and lower bark sizes of the natural rubber tree, as well as the irregularities on its surface. To address these factors, the rubber cutter is first secured to the rubber tree using the adjustable flexible tooth-belt fixing device. This ensures that the center axis of the upper and lower drive teeth stays aligned with the center axis of the rubber tree during the cutting process, enabling precise and stable autonomous execution of the rubber cutting task. When the rubber cutter receives the rubber cutting command, it starts by extending the push rod, which causes the adaptive tree imitation device to stick to the surface of the rubber bark, thereby entering the cutting state. Afterwards, the rubber cutting knife performs a spiral movement along the natural rubber trunk, facilitated by the compound motion transmission device. This movement imitates the cutting trajectory of humans, facilitating efficient rubber cutting operations. After the completion of rubber cutting, the stepping motor drives the rubber cutting device to descend along the screw by a specified distance (tare consumption). This ensures the appropriate consumption of bark for the subsequent cutting cycle. Ultimately, a complete rubber cutting operation is achieved. The entire cutting process can be remotely controlled using an intelligent control module, allowing for the smooth execution of automated cutting operations.

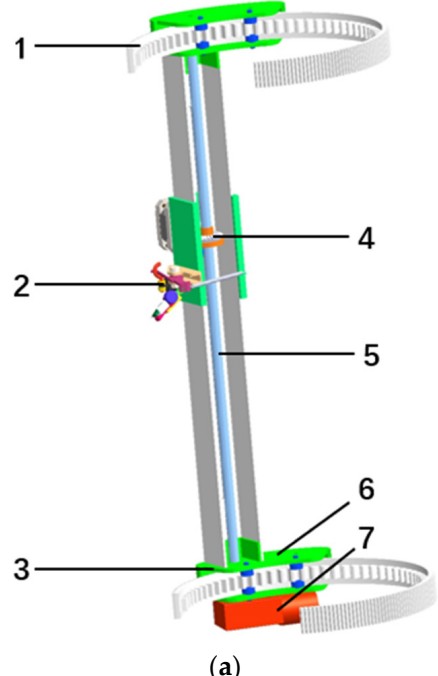
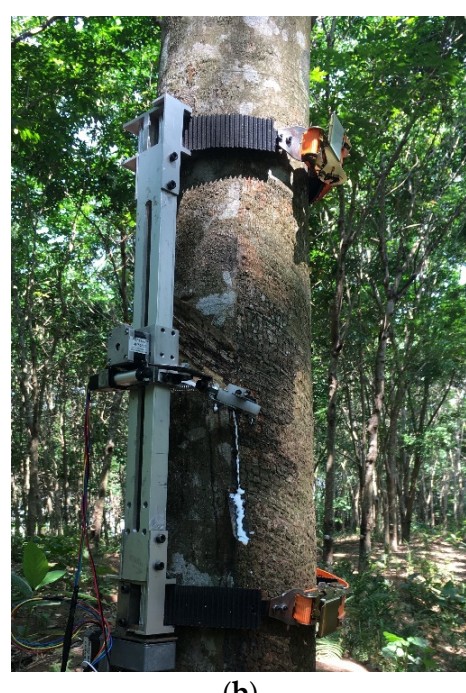

(**a**)　　　　　　　　　　　　　　　(**b**)

**Figure 3.** Intelligent profiling step-by-step automatic glue cutter. (**a**) Three-dimensional structure picture. (**b**) Pictures of Gumlin field work. (1) Adjustable flexible belt fixing device; (2) Adaptive tree-adaptive miter devices; (3) Circular motion device; (4) Vertical motion device; (5) Complete set of screws; (6) Reduction gearing; (7) Motor.

*3.2. Analysis of the FMEA Method for an Intelligent Profiling Progressive Automatic Glue Cutter*

A comprehensive analysis of the failure modes of the intelligent profiling progressive automatic rubber cutter was carried out, as depicted in Table 3.

**Table 3.** Analysis table of FMEA method for intelligent profiling progressive automatic glue cutter.

| Code | Failure Mode | Failure Analysis | Fault Impact | Fault Checking Method | Troubleshooting Measures |
|------|------|------|------|------|------|
| 1 | Motor shaft damage; fracture or deformation | Excessive torque due to overload | Decreased functionality | Regular inspection; instrument testing | Motor replacement |
| 2 | Tooth-belt slippage | Excessive load causes the transmitted force to be greater than the limit of the sum of the frictional forces between the belt and the gear | Decreased functionality | Visual inspection | Increase the belt width or replace the belt |

**Table 3.** *Cont.*

| Code | Failure Mode | Failure Analysis | Fault Impact | Fault Checking Method | Troubleshooting Measures |
|------|-------------|------------------|--------------|----------------------|-------------------------|
| 3 | Blade deformation or breakage | Insufficient blade strength; improper cutting depth and blade installation angle resulting in excessive load causing blade breakage | Loss of function | Visual inspection | Replacement of high strength blades; correction of cutting depth and blade installation angle |
| 4 | Circumferential and vertical movement device rotation is not flexible, there is a jamming phenomenon | The upper and lower tooth-belt gap has foreign matter and installation is not parallel to cause the center axis of the transmission teeth and rubber tree center axis offset; poor lubrication | Decreased functionality | Regular inspections; instrument testing | Removal of foreign objects; enhance lubrication |
| 5 | Unstable amount of skin consumption | The height of the descending screw and the distance from the cutting knife to the guiding depth limiting wheel are not consistent when cutting rubber | Decreased functionality | Regular inspections; instrument testing | Correction of the height of the lowering screw and the distance from the cutting knife to the guiding depth limit wheel |
| 6 | Unstable cutting depth | Improper installation and cutting angle of the blade; spring tension failure, etc., led to jumping of the cutting knife | Decreased functionality | Instrument inspection; visual inspection | Correct blade mounting and cutting angle; replace spring |

*3.3. Analysis of the Traditional FMECA Method for the Intelligent Profiling Progressive Automatic Glue Cutter*

The reliability analysis of the intelligent profiling progressive automatic rubber cutter was conducted using the conventional FMECA method based on the FMEA table.

The RPN is used as a quantitative measure of the hazard, evaluating the potential severity associated with each failure mode. By evaluating the levels of fault occurrence probability, impact severity, and detection difficulty, a comprehensive analysis of the impact is conducted. A higher RPN value indicates a higher hazard associated with the corresponding failure mode, which can be expressed as follows:

$$\text{RPN} = \text{ESR} \times \text{OPR} \times \text{DDR} \tag{14}$$

where ESR represents the level of impact severity, OPR denotes the level of probability of occurrence, and DDR signifies the level of detection difficulty. The ESR is established based on the failure mode, impact, and hazard analysis guide, as well as the maintenance experience of cutter maintenance engineers. The hazard degree values are assigned as

follows: I is assigned a value of 5, II is assigned a value of 4, III is assigned a value of 3, IV is assigned a value of 2, and V is assigned a value of 1. The value of the OPR is determined based on the failure mode, impact, hazard analysis guide, and the available cutter failure data. The possibility of occurrence for each type of hardware failure of the cutter is categorized into five intervals. The corresponding probabilities of fault occurrence $P$ are $P > 10^{-1}$, $10^{-2} < P \leq 10^{-1}$, $10^{-4} < P \leq 10^{-2}$, $10^{-6} < P \leq 10^{-4}$, and $P \leq 10^{-6}$, and the corresponding OPR values are 5, 4, 3, 2, and 1, respectively. The DDR is defined based on various factors, including the precision of the current testing equipment, the expertise of the testing personnel, and the employed testing methods. The assessment of inspection complexity for each component of the cutter involves categorizing them into five types: undetectable, difficult to detect, detectable, easy to detect, and directly identifiable. Correspondingly, the assigned DDR values for these categories are 5, 4, 3, 2, and 1, respectively.

The RPN values for each failure mode of the intelligent profiling progressive automatic rubber cutter are obtained based on the value criteria of ESR, OPR, and DDR. Equation (15) was applied to derive these values and experts from the rubber cutter maintenance industry were invited to provide scores, as illustrated in Table 4.

**Table 4.** Table of RPN values for each failure mode of the intelligent profiling progressive automatic glue cutter.

| Projects | ESR | OPR | DDR | RPN |
|---|---|---|---|---|
| Failure Mode 1 | 4 | 2 | 4 | 32 |
| Failure Mode 2 | 3 | 3 | 2 | 18 |
| Failure Mode 3 | 4 | 3 | 2 | 24 |
| Failure Mode 4 | 2 | 3 | 2 | 12 |
| Failure Mode 5 | 3 | 3 | 3 | 27 |
| Failure Mode 6 | 4 | 3 | 2 | 24 |

Based on Table 4, the hazard levels for the six failure modes can be determined in descending order as follows: Failure Mode 1, Failure Mode 5, Failure Mode 3, Failure Mode 6, Failure Mode 2, and Failure Mode 4. Failure Mode 1 and Failure Mode 5 exhibit the highest RPN values and pose the greatest hazards. Failure Mode 6 and Failure Mode 3 have identical RPN values, indicating the same level of hazard for both failures.

The preparation of the FMEA worksheet, as shown in Table 4, is found to be highly beneficial during the analysis process. By comparing the RPN values before and after implementing maintenance measures, the evaluation of maintenance policies is facilitated. Moreover, a more comprehensive understanding of the failure causes and effects contributes to a better assessment of risk factors.

However, the RPN approach has several shortcomings that have led to the adoption of alternative methods in FMEA. The main criticisms include the following:

(1) Potential result unification: The RPN values may be the same for two different failure modes, despite their values for factors such as ESR, OPR, and DDR being different;

(2) Overlooking significant aspects of failures: In certain cases, failures with high severity may garner inadequate attention due to the low values of other risk factors, resulting in a low RPN value;

(3) Lack of clear distribution pattern: RPNs are distributed from 1 to 125, and the relationship between neighboring numbers at intervals of 5 or 10 relate to each other.

*3.4. Analysis of the Improved FMECA Method for the Intelligent Profiling Progressive Automatic Glue Cutter*

The improved FMECA method is employed to quantitatively analyze the FMEA results and assess the reliability of the intelligent profiling progressive automatic glue cutter. This analysis provides a comprehensive ranking of the hazard level for each failure

mode, enabling the implementation of appropriate measures to promptly address failures and ensure the efficient and reliable operation of the cutter.

The analysis of the intelligent profiling progressive automatic glue cutter within the framework of the FMEA method focuses on evaluating common failure modes through fuzzy synthesis. The evaluation process is outlined in the following steps:

(1) Determination of the factor set: The assessment of fault hazard level in the intelligent profiling progressive automatic glue cutter involves the consideration of the following factors: fault occurrence probability ($u_1$), degree of fault impact ($u_2$), testing difficulty ($u_3$), and fault repair difficulty ($u_4$). These factors collectively form the set $U$;

(2) Establishment of the evaluation set: The primary objective of this study is to identify weaknesses in the intelligent profiling progressive automatic glue cutter and provide guidance for its improvement and maintenance. Since there is no absolute "good" or "bad" state for each influencing factor, an intermediate scale comprising three levels—"better", "average", and "worse"—is introduced between the two extremes. The evaluation levels for the impact factors are categorized into five levels represented by the set **S** = {1, 2, 3, 4, 5}. The specific evaluation levels for each influencing factor are presented in Table 5.

**Table 5.** Evaluation grade table of each influence factor.

| Influencing Factors | Evaluation Level | | | | |
|---|---|---|---|---|---|
| | 1 | 2 | 3 | 4 | 5 |
| Fault occurrence probability $u_1$ | Almost never happens | Rarely happens | Occasional | Sometimes it happens | Frequent |
| Degree of fault impact $u_2$ | Almost no effect | Mild faults | Moderate failure | Critical Failure | Fatal Failure |
| Difficulty of testing $u_3$ | Can be found directly | Easy to detect | Not easy to detect | Hard to detect | Undetectable |
| Difficulty of repairing faults $u_4$ | Simple debugging | Reinstallation | Replacement Parts | Replace the whole machine | Unrepairable |

(3) Determining the fuzzy evaluation matrix involves assigning fuzzy sets to the failure probability, failure impact degree, detection difficulty, and maintenance difficulty of Failure Mode 1. Let the fuzzy set for the failure probability be denoted as $\mathbf{a}_1$ = {0, 0.2, 0.5, 0.3, 0}, the fuzzy set for the failure impact degree as $\mathbf{a}_2$ = {0.1, 0.45, 0.3, 0.15, 0}, the fuzzy set for the detection difficulty as $\mathbf{a}_3$ = {0, 0.15, 0.7, 0.1, 0.05}, and the fuzzy set for the maintenance difficulty as $\mathbf{a}_4$ = {0, 0, 0.5, 0.4, 0.1}. Consequently, the fuzzy evaluation matrix for Failure Mode 1 can be derived as follows:

$$\mathbf{A}_1 = \begin{bmatrix} \mathbf{a}_1 & \mathbf{a}_2 & \mathbf{a}_3 & \mathbf{a}_4 \end{bmatrix}^{\mathrm{T}} = \begin{bmatrix} 0 & 0.2 & 0.5 & 0.3 & 0 \\ 0.1 & 0.45 & 0.3 & 0.15 & 0 \\ 0 & 0.15 & 0.7 & 0.1 & 0.05 \\ 0 & 0 & 0.5 & 0.4 & 0.1 \end{bmatrix} \tag{15}$$

(4) Determining the weight set of influencing factors for Failure Mode 1 involves evaluating the relative importance of these factors in the decision model based on their impact on the decision outcomes. By performing the corresponding calculations, the weight values for each influencing factor are obtained, as presented in Table 6.

**Table 6.** Judgment matrix and weight value of each influencing factor of the plugging device.

| Influencing Factors | $u_1$ | $u_2$ | $u_3$ | $u_4$ | Weighting Value $l_i$ |
|---|---|---|---|---|---|
| $u_1$ | 1 | 5 | 1/3 | 5 | 0.2804 |
| $u_2$ | 1/5 | 1 | 1/7 | 1 | 0.0678 |
| $u_3$ | 3 | 7 | 1 | 7 | 0.5747 |
| $u_4$ | 1/3 | 1 | 1/7 | 1 | 0.0771 |

The judgment matrix can be obtained from the data provided in Table 6 as follows:

$$\mathbf{B} = \begin{bmatrix} 1 & 5 & 1/3 & 5 \\ 1/5 & 1 & 1/7 & 1 \\ 3 & 7 & 1 & 7 \\ 1/3 & 1 & 1/7 & 1 \end{bmatrix} \tag{16}$$

The consistency test of the judgment matrix **B** is conducted based on the value of the consistency ratio $K_C$. Firstly, the corresponding maximum characteristic root $\lambda_{\max} = 4.13$ is calculated from matrix **B**. Substituting it into Equation (8), the consistency index $I_C = 0.043$ is obtained, and from Table 2, $I_T = 0.90$. By substituting these values into Equation (9), $K_C = 0.048 < 0.1$, indicating that the consistency of judgment matrix **B** satisfies the requirements. Therefore, the set of factor weights corresponding to Failure Mode 1 can be obtained as follows:

$$\mathbf{L}_1 = \{0.2804, 0.0678, 0.5747, 0.0771\} \tag{17}$$

(5) Fuzzy integrated evaluation of the cavity rate overload fault. According to Equation (12), the fuzzy comprehensive evaluation vector for fault mode 1 is $\mathbf{D}_1 = \mathbf{L}_1 - \mathbf{A}_1 = [0.3604\ 0.2381\ 0.2962\ 0.1053]$. This indicates that the membership of the cavity rate overload fault to hazard level 1, 2, 3, and 4 is 0.4410, 0.2425, 0.2672, and 0.0493, respectively;

(6) Determining the comprehensive hazard level of excessive cavitation failure. According to Equation (13), the comprehensive hazard level of the cavity rate fault can be obtained as follows:

$$\mathbf{R}_1 = \mathbf{D}_1 \times [1, 2, 3, 4, 5]^{\mathrm{T}} = 3.069 \tag{18}$$

(7) Similarly, the fuzzy evaluation matrices for Failure Modes 2 to 6 are determined as follows:

$$\mathbf{A}_2 = \begin{bmatrix} 0 & 0.6 & 0.3 & 0.1 & 0 \\ 0 & 0.8 & 0.2 & 0 & 0 \\ 0.7 & 0.3 & 0 & 0 & 0 \\ 0.1 & 0.4 & 0.3 & 0.1 & 0.1 \end{bmatrix}$$

$$\mathbf{A}_3 = \begin{bmatrix} 0.1 & 0.3 & 0.3 & 0.2 & 0.1 \\ 0 & 0.1 & 0.2 & 0.3 & 0.4 \\ 0.6 & 0.3 & 0.1 & 0 & 0 \\ 0.1 & 0.2 & 0.7 & 0 & 0 \end{bmatrix}$$

$$\mathbf{A}_4 = \begin{bmatrix} 0 & 0.3 & 0.4 & 0.2 & 0.1 \\ 0.2 & 0.5 & 0.3 & 0 & 0 \\ 0.6 & 0.3 & 0.1 & 0 & 0 \\ 0.5 & 0.2 & 0.3 & 0 & 0 \end{bmatrix}$$

$$\mathbf{A}_5 = \begin{bmatrix} 0.3 & 0.2 & 0.4 & 0.1 & 0 \\ 0 & 0.1 & 0.5 & 0.3 & 0.1 \\ 0 & 0.1 & 0.4 & 0.5 & 0 \\ 0.4 & 0.3 & 0.3 & 0 & 0 \end{bmatrix}$$

$$\mathbf{A}_6 = \begin{bmatrix} 0.2 & 0.1 & 0.5 & 0.2 & 0 \\ 0 & 0.1 & 0.2 & 0.3 & 0.4 \\ 0.3 & 0.6 & 0.1 & 0 & 0 \\ 0.2 & 0.3 & 0.4 & 0.1 & 0 \end{bmatrix}$$

Given that the relative importance of the influencing factors for each failure mode remains consistent, as indicated in Table 5, the same weight set is employed, which is denoted as $\mathbf{L}_1 = \mathbf{L}_2 = \mathbf{L}_3 = \mathbf{L}_4 = \mathbf{L}_5 = \mathbf{L}_6$. Utilizing Equation (12), we can obtain the fuzzy comprehensive evaluation vector for each failure mode as follows:

$$\mathbf{D}_2 = \mathbf{L}_2 \times \mathbf{A}_2 = [0.41, 0.4257, 0.1208, 0.0358, 0.0077]$$

$$\mathbf{D}_3 = \mathbf{L}_3 \times \mathbf{A}_3 = [0.3806, 0.2787, 0.2091, 0.0764, 0.0552]$$

$$\mathbf{D}_4 = \mathbf{L}_4 \times \mathbf{A}_4 = [0.3969, 0.3059, 0.0982, 0.0561, 0.0280]$$

$$\mathbf{D}_5 = \mathbf{L}_5 \times \mathbf{A}_5 = [0.1150, 0.1435, 0.3991, 0.3357, 0.0068]$$

$$\mathbf{D}_6 = \mathbf{L}_6 \times \mathbf{A}_6 = [0.2439, 0.4028, 0.2421, 0.0841, 0.0271]$$

The combined hazard level for each failure mode can be determined using Equation (13):

$$\mathbf{R}_2 = \mathbf{D}_2 \times [1, 2, 3, 4, 5]^T = 1.8055$$

$$\mathbf{R}_3 = \mathbf{D}_3 \times [1, 2, 3, 4, 5]^T = 2.1469$$

$$\mathbf{R}_4 = \mathbf{D}_4 \times [1, 2, 3, 4, 5]^T = 1.6677$$

$$\mathbf{R}_5 = \mathbf{D}_5 \times [1, 2, 3, 4, 5]^T = 2.9761$$

$$\mathbf{R}_6 = \mathbf{D}_6 \times [1, 2, 3, 4, 5]^T = 2.2477$$

Based on the calculated integrated hazard level for each fault, the descending order of hazard levels for the six failure modes is as follows: Failure Mode 1, Failure Mode 5, Failure Mode 6, Failure Mode 3, Failure Mode 2, and Failure Mode 4. After conducting numerous statistical analyses on data from field cutting tests in the forest, the findings demonstrate that the evaluation results are consistent with the real-world scenario, with an accuracy exceeding 95%. Additionally, this method can be applied to analyze the reliability of other agricultural equipment. Additionally, this method enables the analysis of reliability for other agricultural equipment, identification of the most critical potential faults, and implementation of timely measures to mitigate and enhance the design, thus improving the overall reliability of agricultural equipment.

## 4. Discussion

Natural rubber, as one of the four major industrial materials in modern society, exhibits excellent abrasion resistance, impact resistance, elasticity, and heat dissipation. In particular, at low temperatures it demonstrates ductility and resilience that are incomparable to synthetic rubber. It has extensive applications in industrial production, national defense equipment, transportation, medicine, health care, and other fields. Therefore, natural rubber plays a critical role in the national economy and people's livelihoods. Natural rubber is primarily obtained by the semi-spiral ring cutting of rubber trees. In order to

reduce the intensity of artificial rubber cutting and alleviate labor difficulties, the integration of agro-mechanical and agronomic approaches has led to the research and development of intelligent rubber cutting equipment that is replacing the traditional manual rubber cutting. This transition is an inevitable trend. Intelligent rubber cutting equipment significantly reduces the dependence on manual labor, lowers the cost of rubber cutting, and improves the output rate of natural rubber. The reliability of this equipment directly affects the production and quality of natural rubber. The reliability of rubber cutting equipment directly impacts the output and quality of natural rubber. Currently, the failure rate of rubber cutting machines remains significantly high. Once a failure occurs, it greatly reduces the quality of the rubber cuts and may even lead to serious damage to the rubber trees, thereby affecting the income of rubber farmers. Currently, no relevant scientific literature exists on the reliability of intelligent rubber cutters. This paper introduces an innovative method to analyze the reliability of intelligent profiling step-by-step automatic rubber cutters. The original contributions of this paper can be summarized as follows:

(1) Research on the FMECA analysis method based on fuzzy comprehensive evaluation was carried out, and a fuzzy comprehensive evaluation model was established. The model provides a theoretical basis for the reliability design of agricultural equipment and lays the foundation for the application of reliability analysis in the field of agricultural machinery;

(2) A qualitative and quantitative analysis of the reliability of the intelligent profiling progressive automatic glue cutter was carried out by using the improved FMECA method, and the comprehensive hazard ranking of each failure was obtained. We propose improvement measures and formulate a preventive maintenance outline, which can provide a theoretical basis for the improved design of the cutter and check the possible failures of the cutter beforehand, thus reducing the failure rate of the cutter and greatly improving the reliability level of the intelligent profiling progressive automatic cutter;

(3) Taking the intelligent profiling progressive automatic rubber cutter as an example, the above model was used to verify and analyze its frequent failure modes; the results showed that this evaluation result was consistent with the actual situation, indicating that the evaluation model is an effective method that can make full use of the system's fuzzy information for reliability analysis. Furthermore, this innovative agricultural equipment reliability analysis and testing approach holds significant value in elevating the reliability standards of agricultural equipment as a whole and can be explored and implemented in other agricultural machinery contexts.

## 5. Conclusions

This study applies an improved FMECA method to analyze the reliability of an intelligent profiling step-type automatic rubber cutter. The traditional FMECA method suffers from subjectivity and a limited ability for quantitative analysis. To address these issues, we introduced a fuzzy theoretical method in combination with the traditional FMECA method, thus proposing an improved FMECA method based on fuzzy comprehensive judgment. Through this approach, we quantified the qualitative analysis problems, calculated the qualitative analysis results, and analyzed the reliability of the rubber cutter. The study quantifies the problems, calculates the hazard rankings of each failure mode of the rubber cutter, and identifies the areas requiring reliability improvement. It addresses the shortcomings of the original hazard ranking and optimizes the cycle for replacing spare parts, along with the frequency of maintenance and inspection in the field cutting work within the natural rubber forest. This method offers a theoretical basis for the subsequent improvement design of the rubber cutter. Additionally, it enables pre-emptive detection of the potential failures of the rubber cutter, thereby reducing the incidence of failures and enhancing the reliability of the intelligent profiling step-by-step automatic rubber cutter. Moreover, this type of agricultural equipment reliability analysis and detection method exhibits significant technical innovation and positive implications for enhancing the reliability level of agricultural equipment. Furthermore, its applicability can be explored for other agricultural equipment as well.

**Author Contributions:** H.Z.: conceptualization, methodology, investigation, and writing—original draft. Y.C.: investigation and writing—review and editing. J.C.: writing—review and editing. J.L.: writing– review and editing. Z.Z.: writing—review and editing. X.Z.: conceptualization, writing—review and editing, project administration, and funding acquisition. All authors have read and agreed to the published version of the manuscript.

**Funding:** This work was supported by the Key Research and Development Projects of Hainan Province (Grant No.ZDYF2021XDNY198), the Innovation Platform for Academicians of Hainan Province (Grant No.YSPTZX202008, YSPTZX202109), and the National Modern Agricultural Industry Technology System Project (Grant No.CARS-33-JX2).

**Institutional Review Board Statement:** Not applicable.

**Informed Consent Statement:** Not applicable.

**Data Availability Statement:** Not applicable.

**Conflicts of Interest:** The authors declare that there are no conflict of interest.

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
