# Peer review of "Reliability Study of an Intelligent Profiling Progressive Automatic Glue Cutter Based on the Improved FMECA Method"

_agriculture, doi:10.3390/agriculture13081475_

Round 1

Reviewer 1 Report

Although this paper is lengthily written and lack of innovation in methodology, it does present a thorough study on the reliability issues of a natural rubber harvesting machine. The methodology includes failure mode, effect, and criticality analysis (FMECA), expert judgements, risk analysis through risk priority ranking (RPN), fuzzy evaluation, analytical hierarchy process (AHP) in multi-criteria decision making (MCDM) together with mathematics behind them. To this reviewer, the paper looks like a thesis or dissertation of a graduate student and the graduate student did quite a good work. Since, according to aims of the journal, there is no restriction on the maximum length of papers, and the contents of paper are complete, this reviewer considers the paper is a practical application of the above methods and would recommend its publication.

The paper is lengthy, this reviewer just browses but does not have time reading in detail its contents. The followings are his comments and/or suggestion for improvements before publication of the paper.

1. FMECA is the abbreviation of ‘failure mode, effect, and criticality analysis’ but not ‘failure mode, effect, and hazard analysis’.

2. The authors should check the reference quotation format of the journal, especially when a reference is written by two authors.

3. Reference quotation [31] should be [22].

4. Abbreviations such as RPN should be explained when they appear for the first time.

5. Equation typing can be improved. Bold and non-slanted characteristics are usually used for vectors and matrices.

6. Do symbols in Eq. (11) repeat those appearing in Eqs. (3) and (4)?

The English is readable. 

Reviewer 2 Report

Journal: Agriculture (ISSN 2077-0472)

Manuscript ID: agriculture-2525824

Review Report #1

The authors presented an article titled “Reliability study of intelligent profiling progressive automatic glue cutter based on improved FMECA method”. This article falls within the scope of the "Agriculture" journal. However, the article will be ready for publication after a minor revision. Comments are listed below.

1.      What is the novelty of this work? The difference from similar studies in the literature should be explained.

2.      What are the advantages and disadvantages of the FMECA (Failure modes, effect, and criticality analysis) method compared to other methods?

3.      The table 1 should be referenced in the text.

4.      Figure 3 should be defined as "a), b)".

5.      Further discussion can be done in the Results section. Similar studies in the literature can be compared and discussed.

6.      Apparently, conclusions are just observations. The explanations given for the conclusions of the article need to be checked thoroughly.

7.      The article contains numerous typographic and language errors. It should be corrected.

8.      The article should be rearranged by taking into account the journal writing rules and citation rules.
